# Haloarchaea swim slowly for optimal chemotactic efficiency in low nutrient environments

Katie L. Thornton[1], Jaimi K. Butler [2], Seth J. Davis [3,4], Bonnie K. Baxter [2] & Laurence G. Wilson [1✉]

Archaea have evolved to survive in some of the most extreme environments on earth. Life in extreme, nutrient-poor conditions gives the opportunity to probe fundamental energy limitations on movement and response to stimuli, two essential markers of living systems. Here we use three-dimensional holographic microscopy and computer simulations to reveal that halophilic archaea achieve chemotaxis with power requirements one hundred-fold lower than common eubacterial model systems. Their swimming direction is stabilised by their flagella (archaella), enhancing directional persistence in a manner similar to that displayed by eubacteria, albeit with a different motility apparatus. Our experiments and simulations reveal that the cells are capable of slow but deterministic chemotaxis up a chemical gradient, in a biased random walk at the thermodynamic limit.

[1] Department of Physics, University of York, Heslington, York YO10 5DD, UK. [2] Great Salt Lake Institute, Westminster College, 1840 South 1300 East, Salt Lake City, UT 84105, USA. [3] Department of Biology, University of York, Heslington, York YO10 5DD, UK. [4] State Key Laboratory of Crop Stress Biology, School of Life Sciences, Henan University, 475004 Kaifeng, China. ✉email: laurence.wilson@york.ac.uk

Halophilic archaea are ubiquitously slow swimming, in what is presumed to be a response to nutrient-limited conditions[1]. They also possess genetic components for chemotaxis analogous to those found in bacterial species[2,3], suggesting that their chemotactic strategies must be both highly refined, and potentially similar to the 'run and tumble' or 'run and reverse' behaviours seen in eubacteria. However, existing work on halophilic archaeal motility suggests that cells move with long, meandering trajectories, changing direction very rarely[4]. This is inconsistent with current understanding of how Brownian motion limits bacterial chemotaxis[5,6].

The randomising influence of Brownian motion dominates life on the micrometre scale[7] and confounds microorganisms' attempts to navigate. To counter this, bacteria use an intracellular memory of response regulators to integrate stimuli over time[8]. Swimming trajectories are interspersed with quasi-instantaneous changes in direction: tumbles[9], reverse-flicks[10–12], or pauses[13]. The time between re-orientations is modulated by the chemotaxis system, in response to chemical gradients. Meanwhile, rotational Brownian motion randomises a cell's orientation on a time scale $\tau_r = (2D_r)^{-1}$, where $D_r$ is the rotational diffusion coefficient set by cell geometry and fluid friction[7]. This acts as a fundamental limit on chemotactic performance. The prevailing model holds that cells must reorient on time scales shorter than $\tau_r$ otherwise stored information about local chemical gradients becomes invalid[14].

The response regulator CheY plays a central role in many eubacteria. Intriguingly, analogues of this protein play distinct roles in different species: in gram-negative Escherichia coli, an increase in phosphorylated CheY (CheY-P) increases the tumble probability; in gram-positive Bacillus subtilis, increased CheY-P suppresses tumbles[15]. Swimming and taxis in archaea are less understood, although recent work in elucidating the details of the signal transduction network has revealed similarities to bacterial systems[16–18]. Swimming motility in archaea also has some similarities to the bacterial case; for example, quasi-ballistic swimming interspersed with direction reversals has been observed in some species[19]. Recent work on CheY in the halophilic archaeon Haloferax volcanii[3] found structural similarity to CheY in gram-positive bacteria, alongside structural modifications for interfacing with the archaeal motility apparatus.

Here we show that halophilic archaea perform chemotaxis by modulating their run durations in response to a chemical gradient. We measure swimming speeds, reversal rates and rotational diffusivities of cells swimming in three dimensions and use these as inputs to Brownian dynamics simulations. We find that our cells' motility apparatus, their archaella, stabilise swimming trajectories against the influence of Brownian motion, and that their run durations and swimming speed maximise the (fractional) chemotactic drift speed and chemotactic efficiency.

## Results

**Swimming behaviour in three dimensions**. To study chemotactic behaviour in halophilic archaea, we isolated two strains from hypersaline environments: Great Salt Lake (Utah, United States of America (USA))[20] and a desiccated ancient salt lake found underground at Boulby Mine, United Kingdom (UK, Supplementary Note 1)[21]. Our whole-genome sequencing identified the isolates as representatives of the genera Haloarcula and Haloferax, respectively. The strains possess archaeal flagellins (archaellins, Fig. 1), motor complexes and CheY proteins. Cells of both strains were rod-shaped, ~0.7 μm in width, and 1–2 μm in length (Supplementary Fig. 1).

The cells swim slowly, making the discrimination of motile cells from background debris difficult by eye, although swimming cells are clearly visible in a time-lapse image (Fig. 2a). Exponential-phase cultures were imaged using holographic microscopy to record cell trajectories in three dimensions (Fig. 2b, c, Methods section, and Supplementary Movies1–4). Both strains exhibit 'run and reverse' swimming, with average speeds of 1.9 μm s$^{-1}$ ± 0.7 μm s$^{-1}$ and 2.2 μm s$^{-1}$ ± 0.8 μm s$^{-1}$ (mean ± s.d.) for Haloferax sp. Boulby Mine (HXBM) and Haloarcula sp. Great Salt Lake (HGSL), respectively (Fig. 2d, f), with no difference in speed between forward and reverse phases. The distributions of run durations are approximately exponential, with fitted mean values of $\tau_{run} = 14.7$ (12.1) s for HXBM (HGSL), as shown in Fig. 2e, g. Run durations of these halophilic archaea are over 10 times longer than those found in larger bacterial species such as E. coli, and consistent with previous two-dimensional studies of motile halophilic archaea next to sample chamber surfaces[22]. We also note that the relatively low swimming speed of the cells, coupled with the large distances over which we track them mean that the re-orientation events observed take on a V-shaped character in Fig. 2b, c. More detail on the procedure for isolating reversal events (and hence run duration) can be found in the Methods section and Supplementary Fig. 2.

**Rotational diffusion**. To observe the influence of Brownian rotation, we study the directional persistence of our cells during straight runs occurring at least 20 μm away from all boundaries. The rotational diffusivity $D_r$ is found through the direction correlation function, given in the purely Brownian case by $C_B(\tau) = \mathbf{a}(t) \cdot \mathbf{a}(t + \tau) = \exp(-2D_r\tau)$. The cell bodies of both strains undergo small-amplitude helical motion while swimming (Fig. 3a - the dashed line is a guide to the eye showing the helical aspect), complicating the analysis and leading to an overestimate of $D_r$ when this simple model is fitted to the measured direction correlation function. We attribute this phenomenon to the rotational motion of the cell's archaellum[23], driven by a rotary motor[19]. We incorporate a small periodic oscillation into the rotational diffusion to obtain $C(\tau) = \exp(-2D_r\tau)[\cos^2\theta + \sin^2\theta \cos(\omega\tau)]$, where $\theta$ is the helix pitch angle, and $\omega$ is helical angular frequency (Fig. 3b–d, Supplementary Note 2, and Supplementary Eqs. 1 and 2).

Distributions of fitted parameters are shown in Fig. 2e–j, including the key values of rotational diffusivity $D_r = 0.077$ s$^{-1}$ (HGSL) and 0.081 s$^{-1}$ (HXBM), corresponding to $\tau_r = (2D_r)^{-1} = 6.5$ s and 6.17 s, respectively (modal values). These values are similar to those found in Caulobacter crescentus[24], and equivalent to those of an ellipsoid with the same width, but approximately three times the length of the archaeal cells observed. This indicates that although they are thought to have arisen from distinct precursor structures – the type IV pilus in archaea and type III secretion system in eubacteria[18] – the flagellar structures of both archaea and eubacteria increase the effective length of the cells, stabilising them against Brownian rotation[10].

**Chemotaxis experiments**. At low Reynolds numbers, reciprocal motion should not lead to a net displacement of a cell[25]. However, Brownian motion has been shown to break symmetry and enhance the net displacement of such a swimmer[26], a feature that our species appear to exploit. Modulating run duration to achieve chemotaxis is expected to fail when $\tau_{run} \gg \tau_r$[6].

B. subtilis, E. coli, and other model bacterial systems operate in the opposite limit, in which $\tau_{run} \ll \tau_r$. In contrast to this, our strains operate in the ambiguous region where $\tau_{run} \sim \tau_r$. Evidence points to chemotaxis being achieved in species related to ours: capillary and chemical-in-plug assays performed with Haloferax volcanii and Halobacterium salinarum[3,27] resulted in centimetre-scale rings on

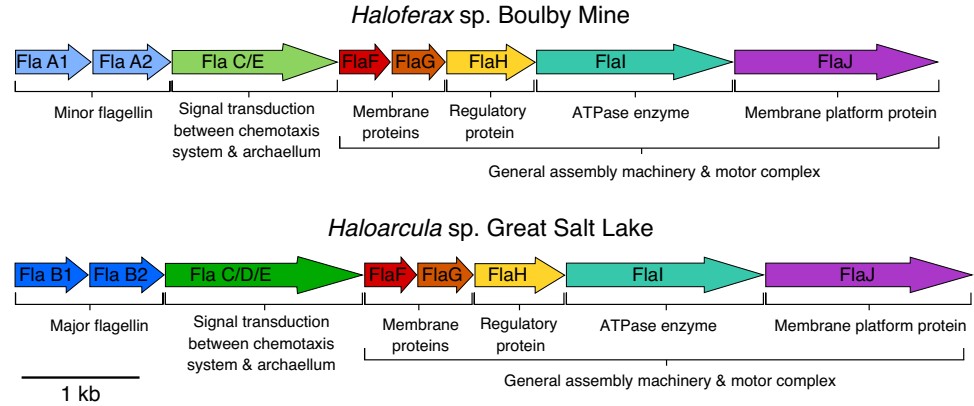

**Fig. 1 Motility gene organisation.** Organisation of the *fla* operons in our environmental isolates *Haloferax* sp. Boulby Mine (HXBM) and *Haloarcula* sp. Great Salt Lake (HGSL) showing flagellins, signal transduction and motor complex components.

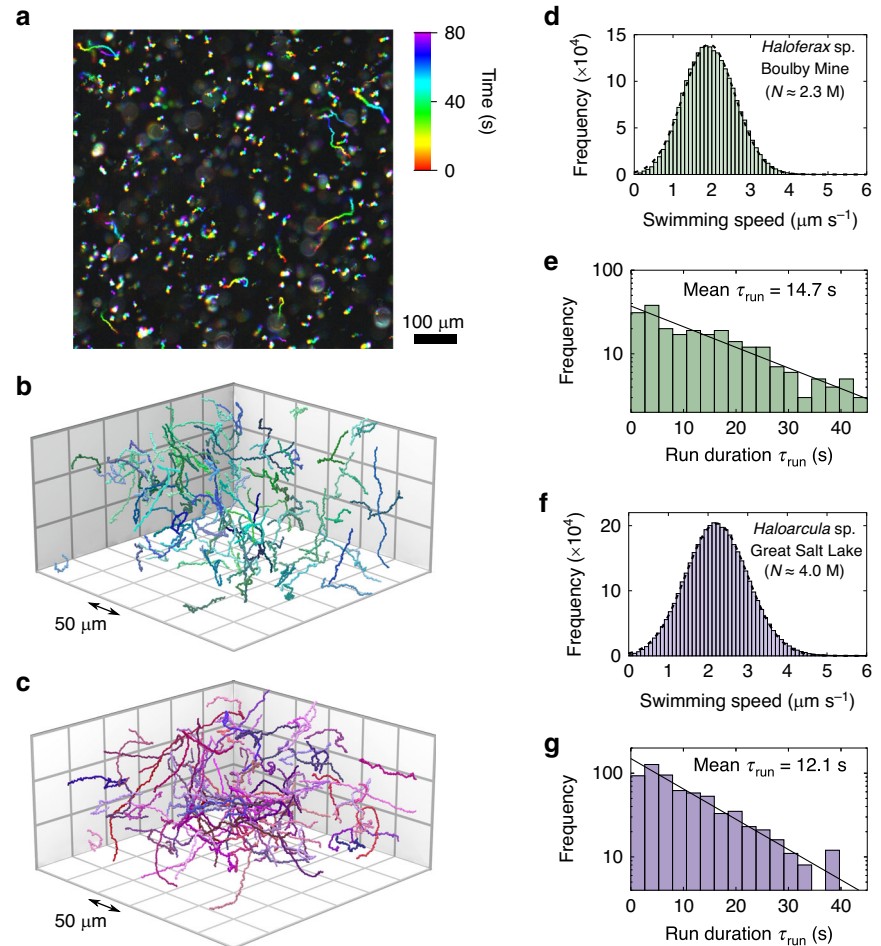

**Fig. 2 3D tracking of extremophile archaea. a** Time-lapse dark field image of *Haloarcula* sp. Great Salt Lake (HGSL) cells. Non-motile cells are small white objects, motile cells are coloured streaks. **b**, **c** Computer rendered three-dimensional tracks of *Haloferax* sp. Boulby Mine (HXBM) and HGSL, respectively, acquired using holographic microscopy. The cells from both strains show characteristic meandering trajectories and infrequent reversal events. **d** Instantaneous swimming speeds for HXBM ($N = 2.3 \times 10^6$ time points), and Gaussian fit (dotted line). The mean swimming speed is 1.9 μm s$^{-1}$ ± 0.7 μm s$^{-1}$ (mean ± s.d.). **e** The distribution of run durations in HXBM is approximately exponential, with a mean run duration of 14.7 s ($N = 232$ runs). **f** Instantaneous speeds for HGSL ($N = 4.0 \times 10^6$ time points), with a mean speed of 2.2 μm s$^{-1}$ ± 0.8 μm s$^{-1}$ (mean ± s.d.). **g** The distribution of run durations in HGSL is approximately exponential with a mean run duration of 12.1 s ($N = 659$ runs).

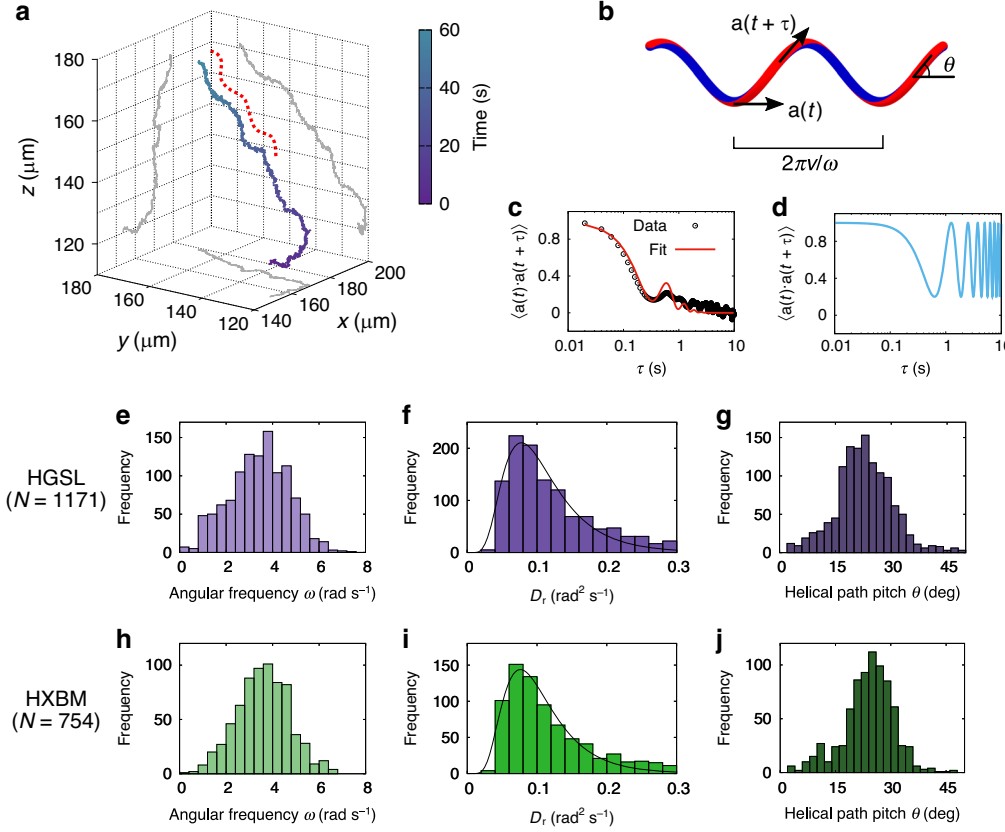

**Fig. 3 Model-based extraction of transport parameters. a** An exemplar trajectory of *Haloferax* sp. Boulby Mine (HXBM), showing helical tendency. The red dashed line is a guide to the eye showing left-handed helix shape, grey lines are planar projections of the track. **b** Idealised helical track showing the parameters in our model: tangent vector (**a**), pitch ($\theta$), and angular frequency ($\omega$). **c** Angular correlation function $\langle a(t) \cdot a(t + \tau) \rangle$ for a single track of *Haloarcula* sp. Great Salt Lake (HGSL). The analytical fit (solid line), shows damped oscillatory behaviour imposed on an exponential decay. **d** Angular correlation of the idealised helix, e.g., in panel **b**, with no rotational Brownian motion. **e–g** Distributions of $\omega$, $D_r$ (including log-normal fit), and $\theta$ of HGSL ($N = 1171$ tracks), extracted from three-dimensional tracks. **h–j** The same parameters for HXBM.

agar plates associated with chemotaxis, although in eubacteria these structures have been seen in the absence of key components of the adaptation system[14], and when chemotaxis or motility is abolished completely[28,29]. Furthermore, maintaining a full complement of chemotaxis genes seems unlikely in the absence of selective pressure to do so.

To investigate further, we performed a chemotaxis assay by filling one end of a sample chamber with methionine-infused agar[27]. Cells were attracted to the region close to the agar interface (a phenomenon not observed in the media-only and saltwater controls, Supplementary Fig. 3). We recorded the three-dimensional swimming dynamics of cells close to the methionine-agar interface using holographic microscopy. The data from all movies were aligned so that the positive $x$ direction points up the chemical gradient. Figure 4a, b show the anisotropic distributions of run durations in the gradient. Cells from both strains showed a significant increase in run duration when swimming up the gradient, compared with runs perpendicular to it. The results are summarised in Table 1, and show that both strains appear to shorten runs slightly when running down the gradient, though this effect is strongest in HXBM and arguably within experimental uncertainties for HGSL. The runs perpendicular to the gradient serve as an experimental control. Chemokinetic effects have been found to enhance chemotactic precision in marine eubacterial systems[30], but there is no evidence that we observe a chemokinetic effect in our archaeal samples: the swimming speed is approximately constant throughout the gradient and constant within individual tracks over long durations (Supplementary

Fig. 4). Examining the difference in $\tau_{\text{run}}$ for cells swimming up versus down the gradient allows us to estimate the fractional chemotactic drift speed[31], $v_x/v_0 = 2[(T^+ - T^-)/(T^+ + T^-)]$, where $v_x$ is the chemotactic drift speed up the gradient and $v_0$ is the swimming speed. Results are shown in Table 1.

**Brownian dynamics simulations**. We explored the fundamental limits of chemotaxis in this context using Brownian dynamics simulations to study how chemotactic drift speed is affected by run duration, reversal rate and swimming speed. If the cells are to climb a chemical gradient efficiently in nutrient-sparse environments, we argue that three conditions that must be satisfied: (i) cells must be able to sense gradients while avoiding saturation of their chemoreceptors (the function of adaptation in eubacteria[32–34]); (ii) they must be able to outpace diffusing molecules in unstable gradients by having a mean-squared displacement that is greater than that for a diffusing molecule. This will always occur if the chemotactic velocity ($v_x$) is >0, but the cells must be able to do so in a time that is shorter than the cell generation time; (iii) in a nutrient-poor environment, cells must minimise the power ($P$) output required to overcome hydrodynamic friction. For a ballistically swimming cell at low Reynolds number, $P = \gamma v_0^2$, where $\gamma$ is a friction coefficient determined by the cell geometry[7]. We use this quantity to define a chemotactic efficiency, $\varepsilon = v_x/P$.

We simulated swimming trajectories with durations in the range $10^4$—$10^6$ s (~3—240 h), orders of magnitude beyond the current spatial and temporal range of single-cell tracking experiments.

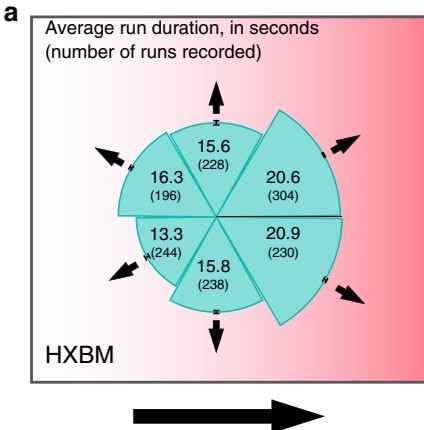

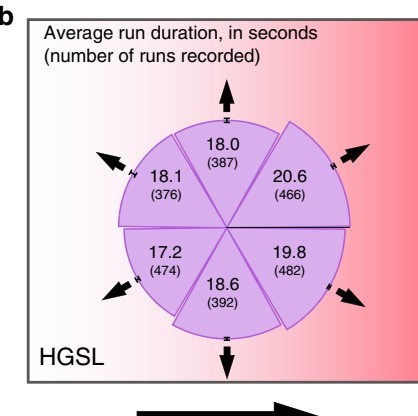

**Fig. 4 Experiments demonstrating chemotactic performance.**
**a** Experimental results for *Haloferax* sp. Boulby Mine (HXBM) moving towards methionine. The sections of the graph show the average run duration in directions relative to the methionine gradient, with the number of runs recorded in parentheses. Error bars on each segment represent s.e.m. (standard error of the mean). **b** Equivalent experimental results for run duration in *Haloarcula* sp. Great Salt Lake (HGSL).

| Table 1 Summary of experimental chemotaxis results. | | | | |
|---|---|---|---|---|
| **Strain** | **Up** | **Perpendicular** | **Down** | **Estimated** $v_x/v_o$ |
| | **Run duration, for directions relative to methionine gradient (s)** | | | |
| HXBM | 20.7 ± 0.3 | 15.7 ± 0.4 | 14.6 ± 0.4 | 0.11 ± 0.02 |
| HGSL | 20.2 ± 0.3 | 18.3 ± 0.4 | 17.6 ± 0.6 | 0.04 ± 0.01 |

These show average run duration up, down or perpendicular to the methionine gradient. The error bars represent s.e.m. (standard error of the mean).

These time scales are also well beyond the doubling time of our strains at optimal conditions (~5 h) and were chosen to allow us to explore the theoretical parameter space fully. Cell–cell interactions were neglected, appropriate to the extremely dilute conditions from which our samples were originally obtained. Parameters such as $\tau_{run}$ and $D_r$ were taken from experimental results (Methods section). Sophisticated models of eubacterial chemotaxis have been demonstrated[35,36], but these are subject to the molecular details of the signalling pathway. These details are not known in our case, so we

implemented a simpler perturbative model in which the reversal probability was modulated in response to the cell's recent history[37,38]. To ensure that cells maintain a linear response (requirement (i) above), the cells's chemotactic sensitivity is chosen to maximise the modulation of the reversal rate, while preventing receptor saturation (see Supplementary Fig. 5 for more details). Figure 5c–e show representative simulated tracks for a non-motile cell (Fig. 5c), an archaeal cell swimming at $v_0 = 2\,\mu m\,s^{-1}$ (Fig. 5d), and an *E. coli* cell (Fig. 5e) swimming at $v_0 = 20\,\mu m\,s^{-1}$. The simulated attractant concentration increases linearly with $x$. The simulated *E. coli* cell shows a comparatively rapid ascent of the gradient, with a chemotactic drift speed $v_x$ approximately five times greater than the archaeon, although the fractional drift speeds $v_x/v_0$ are the same for both motile cells.

In the absence of a gradient, our simulated cells had a constant probability of reversal at each time step, taken from experimental data in Fig. 2. We investigated three possible variants of chemotactic response in which the run duration is modulated: runs shortened in response to a negative stimulus (motion down a chemical gradient), runs lengthened in response to a positive stimulus (motion up a chemical gradient) and a response incorporating both strategies ('bipolar' response). Figure 5d shows how the unstimulated run duration $\tau_{run}$ affects fractional drift velocity ($v_x/v_0$), for a simulated archaeal cell. The swimming speed was set at $v_0 = 2\,\mu m\,s^{-1}$ for all cells in this set. Fractional drift velocity initially increases with run duration, peaking at $\tau_{run} \approx 10$, above which point rotational Brownian motion comes to dominate the swimming dynamics and $v_x/v_0$ decreases. Interestingly, cells that can only shorten their runs perform better when the unstimulated run duration is longer. Our environmental isolates exhibit an unstimulated run duration close to the optimum, while utilising both run shortening and run lengthening to achieve chemotaxis. This adaptation may offer more flexibility in chemical landscapes with a more complicated spatial distribution, or confined environments.

Figure 5e shows the mean-squared displacement per unit time, parallel to the gradient, $\langle x^2(\tau) \rangle / \tau$, as indicated ($N = 100$ simulations per curve). The simulations of *E. coli* were performed using run and tumble statistics obtained from a separate set of three-dimensional tracking experiments on this species (Supplementary Fig. 6). *E. coli* was chosen as a representative bacterium; its run-tumble statistics are similar to those of other species such as *B. subtilis*[39]. At short times, the motile cells show ballistic behaviour: $\langle x^2(\tau) \rangle / \tau \propto \tau$. At intermediate delay times ($10^1 < \tau < 10^2$ s), diffusive behaviour becomes more apparent: $\langle x^2(\tau) \rangle / \tau \propto \tau^a$, where $\alpha < 1$, and $\alpha \to 0$ for the non-chemotactic cells. Chemotactic cells then display a second ballistic regime at the longest times as they move up gradients of diffusing molecules. Example diffusivities for substances with different molecular weights are indicated on the right-hand side of Fig. 5e. All the chemotactic cells can successfully navigate up chemical gradients of the smallest molecules, although the slowest cells require a relatively long time to do so – over 10 h for $v = 1\,\mu m\,s^{-1}$ (considerably beyond the 5-h cell doubling time for our archaeal species). The cells swimming at $2\,\mu m\,s^{-1}$ achieve comparable mean-squared displacements in ~2.5 h, satisfying requirement (ii) above.

**Speed and efficiency.** Finally, we investigated the effect of cell swimming speed ($v_0$) on the fractional drift velocity and efficiency, again using values for $\tau_{run}$ and $D_r$ taken from the experimental data for HXBM shown in Figs. 2e and 3i. Figure 5f shows that fractional chemotactic drift speed ($v_x/v_o$) increases with $v_0$ until $v_0 \approx 2\,\mu m\,s^{-1}$, where it saturates at a value close to that observed experimentally for HXBM ($v_x/v_0 = 0.11 \pm 0.02$). This saturation is not predicted by models like those in Refs. [37,38],

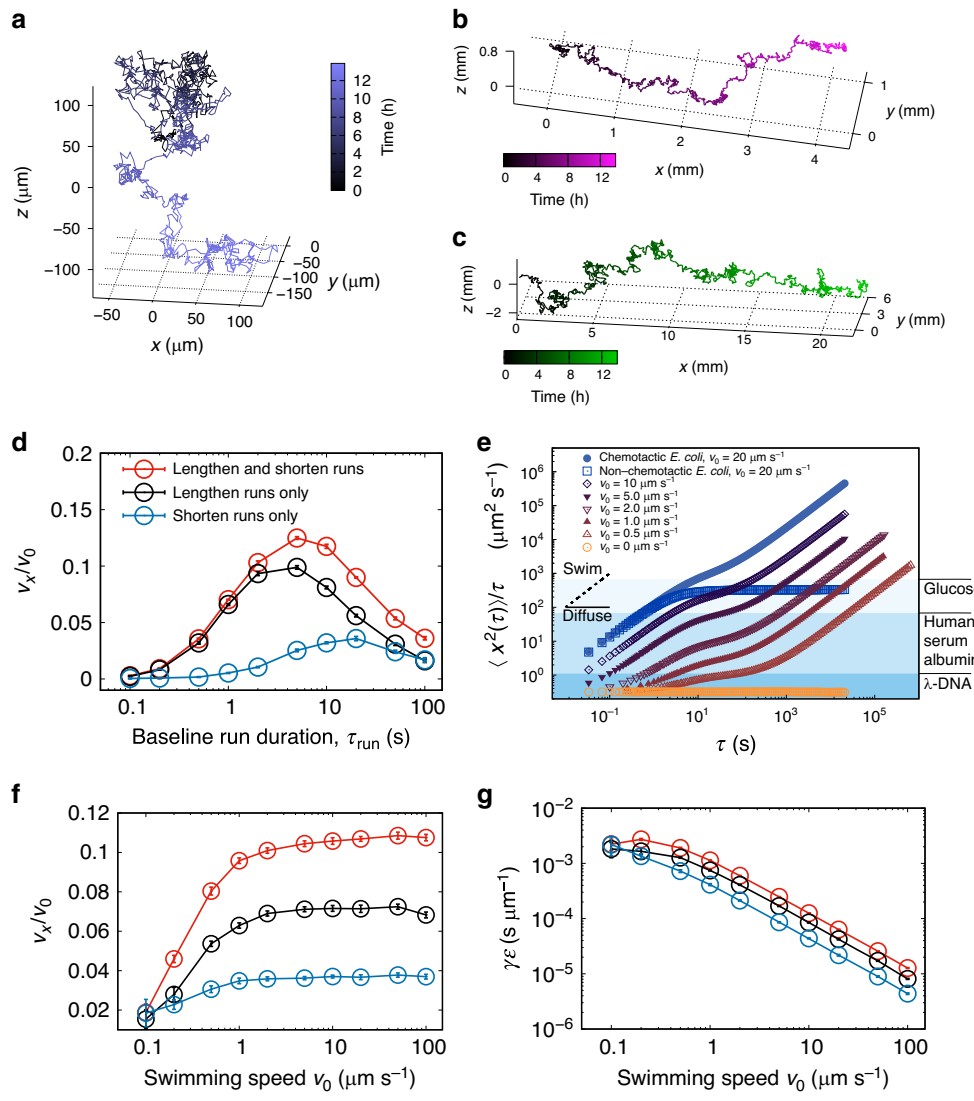

**Fig. 5 Brownian dynamics simulations demonstrating chemotactic performance. a–c** Simulated trajectories of a non-motile cell, a *Haloferax* sp. Boulby Mine (HXBM) cell, and an *E. coli* cell (respectively) in a chemical gradient. **d** Effects of modulating $\tau_{run}$ for chemotactic strategies as indicated. Drift speed increases until $\tau_{run} \sim 10$ s for bipolar and run-lengthening modes. In run-shortening mode, the optimal run duration is around $\tau_{run} \sim 20$ s. Error bars represent s.e.m. (standard error of the mean). **e** Mean-squared displacements per unit time for simulated cells. Series represent HXBM unless otherwise indicated. Molecular diffusivities are given to the right, for comparison. S.e.m. is <0.02% of the values for all but the purely diffusive case and so have been omitted. **f** Fractional chemotactic drift speed of HXBM. The fractional drift speed increases at low $v_0$ before saturating at around $v_x \approx 0.1v_0$. Error bars represent s.e.m., and the series are coloured according to the swimming modes as in panel **d**. **g** Swimming efficiency ($\varepsilon$) multiplied by the friction coefficient ($\gamma$), showing a broad decline in efficiency at higher speeds, consistent with the results in panel **f**. Error bars represent s.e.m., and the series are coloured according to the swimming modes as in panel **d**.

which find $v_x \sim v_0^2$ scaling for the chemotactic drift speed in weak chemotaxis, both in the absence and presence of rotational diffusion (respectively). Rotational diffusion plays a more dominant role in our system compared to that of faster-swimming species like *E. coli*. In our system, $\tau_{run} \gtrsim \tau_r$, so that Brownian rotation is being exploited by our species to randomise swimming direction on a characteristic time scale similar to that of their intrinsic run/reverse dynamics. The nonlinear relationship between $v_0$ and $v_x$ is also manifested in the efficiency which is shown in Fig. 5g, scaled by the friction coefficient $\gamma$. The efficiency is roughly constant across the lower range of $v_0$, and scales as $\varepsilon \sim v_0^{-1}$ for higher swimming speeds. Although our simulations show the slowest-swimming cells performing chemotaxis with the highest efficiency (requirement (iii), above), we reiterate that these cells chemotax too slowly to meet requirement (i). Our strains are capable of

chemotaxis at a considerably higher efficiency than those that swim faster, owing to the onset of $\varepsilon \sim v_0^{-1}$ scaling. As a whole, these simulations indicate that HXBM performs at, or close to, the optimum theoretical chemotactic efficiency. The performance of HGSL in experiments is further from the optimum efficiency defined by our linear model (though approximately the same magnitude). There could be several explanations for this discrepancy, for instance that the model might not be as applicable to this strain due to details of the adaptation pathway, or that methionine is simply a weaker attractant to this species.

In conclusion, we have studied motile environmental isolates of halophilic archaea using three-dimensional tracking and Brownian dynamics simulations, and found their chemotactic efficiency to be optimal. Though the chemotactic system controlling reversal probability is structurally similar to that

found in bacteria, our microorganisms have different motility structures[40]. Our study sought to characterise this distinctive system. Counterintuitively, given their very long run durations, these cells are capable of slow but efficient chemotaxis. We speculate that this is an adaptation to an environment in which nutrients are scarce and competition is extremely limited, as it constitutes a minimum energy consumption strategy for achieving chemotaxis by swimming. In contrast to this behaviour, many of the most widely studied motile eubacteria exist where both nutrients and competition are more abundant, requiring them to swim (and consequently consume nutrients) much faster. These differing eubacterial and archaeal strategies demonstrate the general utility and adaptability of flagellar (archaellar) motility in different conditions, mediated by a change in swimming dynamics. Finally, we note that our results are a limiting case, namely chemotaxis in an isotropic fluid. Cells in a structured environment, such as rock or salt crystals with micrometre-scale fissures, would have their rotational diffusion inhibited by the walls of their local environment, preventing them from rotating about their short axes. Such confinement could increase $\tau_r$ substantially, extending the cells' directional persistence.

## Methods

**Cell culture for motility experiments**. Archaeal cells were cultured in the standard laboratory archaeal medium Modified Growth Media (MGM)[41] at 25% dissolved salt, adjusted to pH 7.5. When preparing the growth medium Oxoid-Peptone was used as other peptones have been shown to contain bile salts which lyse haloarchaeal cells[42]. For each sample, 10 ml of media containing 50 μl of saturated culture was incubated at 45 °C in an orbital shaker at 150 rpm for 18 h to early exponential growth phase. Three to five biological replicates were made for each experimental condition, each of which was imaged in at least five movies. Sample chambers were constructed from glass slides and cover slips, with a sample volume measuring ~20 × 10 × 0.4 mm$^3$ and loaded by capillary action.

**Genome sequencing**. DNA was extracted from the archaeal samples using the technique of ethanol precipitation recommended for halophiles[43] with STE buffer used in place of phenol. Deionised water was used to rapidly lyse the cells without damaging the genetic material. DNA was isolated and purified using a NucleoSpin PCR clean-up kit. DNA quality and concentration were assessed using gel electrophoresis and micro-volume spectrophotometry (NanoDrop). DNA sequencing was outsourced to a specialist prokaryotic genetics laboratory (Microbes NG, University of Birmingham, UK). Visualisation of the whole genome was undertaken using Snap Gene Viewer (version 4.1.9) open source software. Genetic data was then compared with published genes using the BLAST database[44]. Exact species were difficult to reliably deduce due to the high levels of genetic transfer typically associated with haloarchaea[45].

**Digital holographic microscopy**. Our cell tracking experiments were performed using digital holographic microscopy, as described elsewhere[12,46–48]. The samples were imaged on a Nikon Ti inverted microscope. The illumination source was a Thorlabs single-mode fibre-coupled laser diode with peak emission at 642 nm. The end of the fibre was mounted above the specimen stage using a custom adaptor and delivered a total of 15 mW of optical power to the sample. A Mikrotron MC-1362 monochrome camera was used to acquire videos of 3000 frames, at a frame rate of 50 Hz and with an exposure time of 100 μs. A ×20 magnification bright field lens with numerical aperture of 0.5 was used to acquire data at a video resolution of 512 × 512 pixels$^2$, corresponding to a field of view measuring 360 × 360 μm$^2$. The raw videos were saved as uncompressed, 8-bit AVI files.

**Holographic data reconstruction**. Each individual video frame was used to calculate a further 150 images corresponding to a series of slices throughout the sample volume. The images were spaced at 2 μm apart along the optical axis and in total constituted a reconstructed volume of 360 × 360 × 300 μm$^3$. The stack of numerically refocused images was calculated using the Rayleigh-Sommerfeld back-propagation scheme[49]. We localised the cells[50] using a method based on the Gouy phase anomaly, described in more detail elsewhere[46,51]. This method segments features based on axial optical intensity gradients within a sample, allowing us to extract 3D coordinates for individual cells in each frame. Lateral position uncertainties were ~0.4 μm, while the axial performance was slightly worse at ~0.5 μm. The latter was limited by the angular resolution of the microscope objective. A separate software routine was used to identify which coordinates in subsequent frames correspond to the same cell, based on their proximity. These coordinates were arranged into tracks and smoothed using piecewise cubic splines in order to remove noise in the cell coordinates and improve estimates of cell velocity as described in previous work[12]. Examining the mean-squared displacement of the cells' smoothed trajectories allowed us to discriminate between swimming and diffusing cells, and to discard the latter. The smoothing process also allowed for linear interpolation of missing data points, up to 5 points (equivalent to 0.1 s). Tracks with a duration shorter than 3 s, typically cells entering or leaving the field of view, were discarded. To identify points where the swimming cells reverse direction, we define a heuristic measure $\Xi$ based on the cell's swimming speed and angular speed (the rate at which the cell is changing direction):

$$\Xi(t) = \frac{|\mathbf{a}(t) \cdot \mathbf{a}(t+1)|}{\Delta t} \cdot \left[1 - \frac{v_t}{\langle v \rangle_t}\right], \tag{1}$$

where $\Delta t$ is the time between successive position measurements in a track, $v_t$ is a cell's instantaneous speed and $v_t$ is the average speed for the whole track. Example data and values for $\Xi$ are given in Supplementary Fig. 2.

**Numerical simulations of archaeal cells**. We used the values for swimming speed, run length and rotational diffusivity obtained from *Haloferax* sp. Boulby Mine to set up Brownian dynamics simulations. Cells were modelled as prolate ellipsoids, subject to rotational and translational diffusion[7,52]. The viscosity of the growth medium was measured to be 1.82 ± 0.01 cp using a concentric-cylinder rheometer. The ellipsoids move along their major axis with a fixed speed, and a probability of reversing drawn from the distribution fitted to experiments (Fig. 2e), with a mean run time of 14.7 s. The position of the centre of mass of all cells in each run $\mathbf{r}_{avg}(t) = (1/N) \sum_{i=1}^{N} \mathbf{r}_i(t)$ was recorded as a function of time. A straight-line fit through the origin was performed on these data sets to extract values for $\mathbf{v}$, the drift velocities: $\mathbf{r}_{avg}(t) = \mathbf{v}t$, where $\mathbf{v} = (v_x, v_y, v_z)$ and $\langle v_x \rangle$ is the drift in response to the gradient. The parameters used in simulations (unless specified) were those obtained in experiments. To summarise: for the model archaea, $\tau_{run} = 14.7$ s, $D_r = 0.08$ s$^{-1}$, $v_0 = 2$ μm s$^{-1}$ and cells reverse only. For *E. coli*, $\tau_{run} = 1$ s, $D_r = 0.08$ s$^{-1}$, $v_0 = 20$ μm s$^{-1}$, and the tumble angles are drawn randomly from the experimentally derived distribution in Supplementary Fig. 6. *E. coli* cells are permitted to lengthen their runs only, in line with experimental evidence[9].

**Chemotaxis model**. To simulate chemotaxis, we biased a cell's tumble probability depending on whether it has moved up or down a simulated chemical gradient. For simplicity, the chemical concentration was given by $c(x,y,z) = x$, i.e. a constant gradient in the positive $x$-direction. Motivated by the analogy to eubacterial two-component signalling, we adopt a general approach[38,53,54] to calculate a time-dependent tumble rate $\lambda(t) = \lambda \left[1 - \int_{-\infty}^{t} dt' c(t') R(t-t')\right]$, where $\lambda = 1/\tau_{run}$, and $c(t')$ is the concentration experienced by the cell at time $t'$. The cell's chemical response function is given by $R(t) = W k e^{-kt}[1 - kt/2 - (kt/2)^2]$, where $W$ is the chemotactic sensitivity and $k$ is a rate constant that describes how long a cell retains information about previous chemical concentrations to which it has been exposed (i.e. the length of its 'chemical memory'). We tuned two parameters: the unbiased tumble rate $\lambda$ and the swimming speed $v$. The quantity $W$ was chosen to ensure the maximum chemotactic response while minimising saturation (see Supplementary Figs. 5 and 7 for more details). Previous studies[37,38,54] have coupled $R(t)$ to the base tumbling rate such that $\lambda = k$, in line with experiments[14]. we explore the effect of decoupling $\lambda$ and $k$ in the extended [...] data and choose a memory length of $1/k = 2$ s. Simulations were performed for batches of 100 cells under each condition, simulating cells swimming for $10^4$ s with time steps of 0.033 s, except in the case of the mean-squared displacement results (Fig. 5e), in which we conducted simulations for much longer time scales, up to $3 \times 10^6$ s, in order to collect sufficient statistics at long times. Initial swimming directions were chosen randomly by allowing the direction vector to diffuse randomly for 60 s before the start of each trajectory.

**Chemotaxis experiments**. The verify our results, we performed three-dimensional tracking experiments on both environmental isolates in a chemical gradient of methionine, which is known to be an attractant to other halophilic archaea[24]. The gradient was formed by preparing saltwater agar in liquid form at 50 °C, and adding methionine to a final concentration of 5 mM. This liquid media was quickly pipetted into the end of a sample chamber where it cooled to form a solid plug. When the agar had cooled sufficiently, cell suspensions were pipetted into the other end of the chamber, which was then sealed. The chamber was left for around an hour for the gradient to become established and a band of cells to form close to the agar interface. Cells within the band, ~1–2 mm from the interface, were imaged as in previous experiments.

## Data availability
The tracking and simulation data supporting this study are available at the York Research Database (https://doi.org/10.15124/6b44e8e4-a22c-4a55-a2fc-b2976a06d2b4). Sequence data that support the findings of this study have been deposited in the European Nucleotide Archive with the primary accession code PRJEB33805. All materials are available from the authors upon reasonable request.

## Code availability

Details of the holographic reconstruction routines have been published elsewhere[49], and strategies for three-dimensional cell tracking have been published in the corresponding references[46,51]. The simulation scheme has been published by previous authors[38,54] and we have used the parameters given in the text.

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

## Acknowledgements

We thank M.A. Bees and L. Turner for helpful discussions. We thank the teams at STFC Boulby Underground Laboratory and the ICL-UK Boulby mine for access to the site and support in work carried out, and C.S. Cockell at the Mine Analogue Research (MINAR) at the University of Edinburgh for logistical support. Genome sequencing was provided by MicrobesNG (http://www.microbesng.uk), which is supported by the BBSRC (grant number BB/L024209/1). The authors would like to acknowledge support from the Rowland Institute at Harvard and EPSRC grant EP/N014731/1 (to L.G.W.), the W.M. Keck Foundation (to J.K.B and B.K.B.) and the 111 Project grant D16014 (to S.J.D.)

## Author contributions

L.G.W., K.L.T. and B.K.B. designed the research, L.G.W., K.L.T. and J.K.B. conducted the experiments and simulations. L.G.W., K.L.T. and S.J.D. interpreted the data. L.G.W. and K.L.T. wrote the manuscript and all authors commented on it.

## Competing interests

The authors declare no competing interests.
