## [Peer Review File · Nature Communications]

Reviewers' comments:

Reviewer #1 (Remarks to the Author):

In their manuscript, Thornton et al. investigate swimming motility of two halophilic archaea natural isolates. In general, archaeal motility and chemotaxis is a very interesting and poorly investigated topic, which certainly merits more attention. These organisms swim very slowly compared to bacteria, making analysis of their motility challenging. Using state-of-the-art holographic microscopy, the authors have been able to characterize the long-term random motion of the archaea. They found that archaea swim using a run-reverse mode of locomotion, with a swimming speed of approx. $2 \mu\text{m/s}$ (10 times slower than *E. coli*) and run time in the order of 10-15 s (nearly 10 times longer than *E. coli*). The authors also extracted other swimming parameters, including rotational diffusion, which had comparable timescale to the run time. They then performed numerical simulations to evaluate the ability of the archaea to chemotax in a gradient of attractant, assuming that the internal chemotaxis pathway dynamics is similar to the reference pathway of *E. coli*. Based on these simulations, they argue that, given their swimming velocity and run duration, archaea should be capable of performing chemotaxis, and that their low swimming speed maximises the energetic efficiency of chemotaxis.

Although the study addresses a very interesting topic and is technically sound, current level of experimental support is not sufficient, and one of the key authors' arguments is not well justified. I believe that additional experimental work will be required to warrant publication at high level.

Major points:

1. A major gap in the manuscript is the lack of an experimental measurement and characterization of the archaeal chemotactic response. Although this manuscript focuses on chemotaxis, only motility is analysed experimentally, whereas all conclusions related to chemotaxis rely on computer simulations. This significantly limits the impact of this manuscript. Given authors' expertise in studying microbial motility, I believe that such experimental analysis should be feasible. Moreover, it would provide essential parameters for the modelling and simulations of the pathway behaviour (at the minimum, choice of k and discrimination between scenarios of chemotaxis strategy).
2. There is no clear rationale given for quantifying efficiency of chemotaxis (in other words the cost/benefit balance) as v_{drift}/v^3 . The higher the chosen power of v_0 , the more advantageous low swimming speed will appear, so this choice must be carefully justified. Since the dissipated power scales as v_0^2 , as the authors note, why not choosing v_{drift}/v_0^2 as the efficiency quantifier? It seems that v_{drift}/v^3 scales roughly as $1/v_0$, so that v_{drift}/v_0^2 would be more or less constant as a function of v_0 . Since the authors claim that archaea are most energy efficient at their chemotaxis relies heavily on this choice of v_{drift}/v^3 , it is not well founded in my opinion.

Minor points:

3. Simulations: It should be made clearer how v_{drift} (but also other quantities) is evaluated. Is it simply $(x_{\text{final}} - x_{\text{init}})/T_{\text{simu}}$? The authors also use v_x , is it the same as v_{drift} ? If not, how is it different from v_{drift} ? The same goes for v_0 and $|v|$.
4. Simulations: Fig. 3g,h: Above a run length of 10 s, the authors note that the average drift is noisy, but do not comment on this. Do they have an explanation for this? Is it possible that the limited number of simulated cells (200) is playing a role?
5. Fig. 3 d-h: please provide error bars, here and in Fig. S2.
6. Experimental: What were the criteria to distinguish tumbles/reorientations from runs when estimating run length? Please include them explicitly, at least in supplementary.

Reviewer #2 (Remarks to the Author):

Thornton, et al. report on swimming behavior of environmental isolates of halophilic archaea, and provide a numerical simulation of the motility based on the parameters extracted from a holographic microscopy. Overall, I found this work interesting in the field of cell motility in archaea. However, several important experiments and inconsistencies should be properly addressed before to recommend this manuscript for publication. I have several comments, which somewhat exaggerates the novelty of the conclusions.

Comments

Do the authors have a direct evidence that two strains of HXBM and HGSL swim slowly in low-nutrient environment? The title of the manuscript says low-nutrient, but the method described that the cell behavior was observed at early exponential phase, that the growth condition is rather rich-nutrient. Please provide additional data how cells move in low-nutrient environment or change phrases.

I do not fully understand what is the reason that the author studied environmental isolates halophilic archaea, rather than the model species of *H. salinarum* or *H. volcanii*. Is there any evidence that two strains of HXBM and HGSL have chemotaxis or other movements in response to stimuli such as oxygen or light? Please clarify the significance of two species.

I don't see why Fig. 1a is important for the paper. The authors presented only fla operons, but several transducer proteins also known to be involved in taxis as listed in a recent review (<https://doi.org/10.1042/ETLS20180089>). Please refer more extensively to published work, and clarify which type of taxis is known to have in two species.

The authors state that the data in Fig. 1cd show that 'run and reverse' swimming, but I am not convinced by the data shown. How do authors distinguished the cell behavior of run and reverse, and measured the run durations? The author should provide additional data clearly demonstrating the data extraction of the run and reverse durations.

Given the low resolution of image in Fig. 2a, the author should provide additional data clearly demonstrating the helical tendency of the trajectory that shows the handedness of cell body precession.

Reviewer #3 (Remarks to the Author):

The authors report analysis of three-dimensional trajectories of two halophile archaea, which are notable for how slowly they swim. Run-reverse motility was observed with extremely long run times. Rotational diffusion was measured by analysing the drift in direction of the small amplitude helical swimming during runs, with rotational diffusion reorientation timescales similar to *E. coli* and *C. crescentus*, corresponding to the combined rotational drag of the combined body and flagellum, as in those species. Numerical studies using Brownian dynamic simulations of run-reverse motility and slow swimmers showed that a population of cells could drift along a favorable chemical concentration gradient despite the relatively larger effects of rotational diffusion for slow swimming. Notably, in the simulations the efficiency of chemotaxis (using a specific metric of efficiency, discussed below) is increased since the power required to swim at slow speeds is much reduced.

My overall suggestion is not to publish the manuscript in Nature communications. While the experimental work provides definite new information for these two species of archaea, the biophysical results (i.e., size

rotational diffusion constant) are not much unexpected. In addition, the Brownian dynamics chemotaxis results, while of some interest (though see my comment below) doesn't really seem connected to the experiments -- indeed, the authors do not even show experimentally that these species can chemotax. They only argue (top of p3) that related species have shown somewhat weak results using agar plate assays, and that the presence of chemotaxis genes implies that these species must chemotax. The simulations thus seem to answer a question not directly related to the experimental findings, and there are no experiments done to probe any of the points raised by the simulations.

Comments:

1) As mentioned above, a serious weakness of the paper is that chemotaxis is never demonstrated for these species, hence the investigation using simulations seems only tangentially related.

2) The result that the flagella likely decrease rotational diffusivity is not really new, since the same physics is observed in E coli (as the authors say and reference relevant papers).

3) The choice of efficiency metric seems to need some more justification. In particular, I wonder whether looking at a metric of raw drift velocity over power for swimming is more biologically relevant. ($\sim v_{\text{drift}}/v_0^2$). With one less power of the swimming speed in the denominator, the major conclusion that such slow swimming as observed (2 micron/s) is most efficient is likely no longer accurate.

Thornton et al., response to referees

Referee #1

We would like to thank the referee for their broadly positive response, and address their specific points:

1. The referee notes that there is no experimental validation of our simulations of chemotactic performance, and asks for this and some of the resulting parameters.

We have now performed additional experiments demonstrating the chemotactic performance of both of our lab strains. We discuss the chemotactic strategy of the strains and give values for the biased run durations both up the gradient (significantly longer) and down the gradient (apparently shorter, but a statistically weaker result). Although we cannot directly determine the length of the cells' memory using our approach (i.e. determine the value of $1/k$), our simulations indicate that a memory length longer than the average run duration degrades performance and would result in a more isotropic distribution of run durations. This allows us to propose an upper bound of τ_{run} on the value of $1/k$ (from experiments), and to suggest that the true value of $1/k$ lies in the range 1-5 seconds (from simulations).

2. The lack of a rationale behind the efficiency metric v_{drift}/v_0^3 .

We can see the point made by this reviewer and reviewer #3 (please see below). The choice of normalising the *fractional* drift velocity by swimming power was made under the assumption that it is not surprising that faster-swimming cells might chemotax faster. It would therefore seem reasonable that a relative measurement of drift speed might make sense. Given the improved quality of data, we have refined the argument. Our strains swim fast enough to give them an advantage over non-swimming cells, justifying the (energetically costly) maintenance of swimming apparatus. They are also capable of chemotaxis, which we now show experimentally. Nevertheless, their swimming speed is right at the bottom of the range for which the above statements are true: the minimal power consumption condition. This low power demand is ideally suited to a low-nutrient environment in which encounters with metabolizable molecules are infrequent.

Minor points:

3. The referee has asked for clarification of the quantities used, and how they are defined.

This has been amended as indicated in the manuscript.

4. The referee asks whether small sample size might account for some uncertainty in the results

This has been amended and clarified with a significantly (10x) larger set of simulation data. The noise in the data has been significantly reduced, but the overall trends for the 'optimal' parameter choices are unaffected and our conclusions are unchanged.

5. The referee has asked for error bars on the simulation figures in the main manuscript and supplementary material

These have been added where appropriate. We have chosen not to add error bars to the 'directionality' panel Fig. 3g because it is a ratio of quantities dominated by the chemotactic drift

v_x . The ratio is equal to 1 over most of the range and the perpendicular velocities (v_y , v_z) tend to zero, giving poorly conditioned/misleading numbers when combining errors in the standard way.

6. *The referee has asked for the criteria we used to determine reorientations*

This has been explained in detail in the supplementary material, along with an example data set.

Referee #2

We would like to thank the referee for their thoughtful and constructive responses, and address their specific points:

7. *The referee asks for confirmation that our strains swim slowly in low-nutrient environments.*

We have added more calibration data for cells suspended in saltwater media in the supplementary information, noting that the swimming behaviour is largely unchanged.

8. *The referee has asked why we chose to study two environmental isolates rather than model lab strains, and whether our strains demonstrate movement in response to light or other stimulus*

The choice to use environmental strains was really made from a sense of curiosity; we had access to some interesting sites for sampling and we took advantage of these. It was intriguing to us that our two strains have completely different environments in terms of ambient illumination (harsh UV in the Great Salt Lake; complete darkness in the mine) and yet show similar swimming phenotype, suggesting that aspects of their behaviour might be shaped by what is common to both. We have now demonstrated chemotaxis in both strains in our new experiments (please see point 1 above).

9. *The referee has queried the purpose of Fig. 1a, and asked us to include an additional references.*

Fig. 1a (along with the genetic data in the ENA) establishes that our strains have a relatively conventional set of archaeal flagellar genes, which reinforces our interpretation of the motility as driven by flagella. We agree that transducer proteins are likely to be critical to the internal signalling pathway (we have included the reference suggested by the referee, along with two other salient works for more context), but our study focuses on the impact on motility rather than the internal workings of the cell.

10. *The referee asks how we know that our cells demonstrate a run/reverse phenotype, and how we determine run duration.*

We thank the referee for their suggestion and have included new data in the supplementary information (Fig. S3), along with an example data set, to demonstrate how we isolate reversals – essentially a combination of a change in direction with a reduction in speed. We also note that because the cells swim slowly and our tracks are taken over a large spatial range, the cells don't appear to precisely re-trace their steps in a reversal event. This is due to Brownian rotation continuously changing the orientation of the cell during swimming, and results in reversals that have a more 'V-shaped' character. We have added a sentence to the manuscript to this effect.

11. *The referee has asked for a higher-resolution version of Fig. 2a, showing the helical character of the track.*

We have attempted to highlight the helical aspect of the tracks that the referee describes but note that the 'noise' in the cell track position is physical and due to both translational and rotational Brownian motion rather than an artefact due to noise in the image acquisition system. The clearest indication of the helical nature of the track consequently comes from *statistical averages* such as the angular correlation function in figure 2c, which we use to observe the averaged signal amid the positional and orientational noise. In the case of a purely Brownian deviation in direction, this function would drop monotonically to zero. The oscillations in Fig. 2c (black data points) trace a damped cosine function that is characteristic of helical motion in the presence of orientational noise. When a simple (non-oscillating) model is used to fit the data, the rotational diffusion coefficient is significantly overestimated; when this erroneous value is used in simulations, the (simulated) tracks do not look correct, hence the more refined model. We have included a guide to the eye on Figure 2a, reduced the data point size and amended the figure caption to aid with the visualisation of the helical aspect of the track.

Referee #3

We thank the referee for their thoughtful and constructive comments on our manuscript and would like to address their specific criticisms. Some of these recapitulate the concerns of previous reviewers, and the central criticism surrounds the lack of an experimental component to match the simulations; we acknowledge this (the other reviewers highlight the issue as well) and we have addressed this point in our responses above, performing new experiments. This evidence shows that our strains do indeed perform chemotaxis and do so by modifying their run length.

12. The referee mentions that while the information on the archaea is new, it is not unexpected

Although we disagree strongly with this comment, the comment is very helpful because we feel that it highlights an under-emphasised aspect of our manuscript. We note first that caution should be applied when assuming things about archaeal systems based on bacterial results (in the same way as one would for eukaryotes). Archaea have a completely different flagellar system to bacteria, and the current understanding in the field is that it evolved from a distinct origin (type IV pilus, as opposed to the type III secretion system in bacteria). The fact that the resulting phenotype is similar - rotary motors, stabilisation against Brownian rotation – is an intriguing aspect of convergent evolution. Nature has found the same solution by two distinct pathways. We have now highlighted aspect this in the main manuscript.

13. The referee queries the use of the efficiency metric

The first reviewer also commented on this; as in our response to point 2 (above), we have modified our claim, and the title of the manuscript. We still maintain that we have demonstrated that these species are optimally adapted to low-nutrient environments. Moreover, we maintain that the results that we observe – the first 3D tracking of archaea and a new and distinct chemotactic strategy that runs counter to previous understanding of life at microscopic scales – are novel and remarkable.

Reviewers' comments:

Reviewer #1 (Remarks to the Author):

In this revised version of their manuscript, the authors have addressed several issues raised by other referees and myself. Most importantly, they have now provided experimental evidence that the two studied species of archaea indeed show chemotaxis.

However, as the manuscript stands now, these new experiments remain disconnected from simulations. While the authors show that cell run duration depends on the direction along the gradient, they do not give the value for the chemotactic drift. This is rather odd, since it is the drift that they analyse in their simulations. Critically, given the substantial difference between the run duration up or down the gradient, one could expect the drift value to be rather high, on the order of several percent. This would be in contrast to the very small drift predicted by simulation (below 0.2 % in Fig. 3f). Understanding this difference between the simulations and experiments would be essential.

The average run duration seems to increase in presence of a gradient, even for runs down the gradient, compared to medium without attractant. Is this a signalling or a metabolic effect?

Moreover, the arguments supporting the central claim of the manuscript, namely that haloarchaea have optimised energetics of chemotaxis in low-nutrient environments, have been largely removed from the manuscript. They are now replaced by a general argument that swimming at low speed is energetically less costly. While this is true, it is also rather obvious. And since it is established that slowly swimming archaea can chemotaxis, such general statement does not really add much to what is already known.

Reviewer #2 (Remarks to the Author):

The authors did an excellent job in improving their manuscript and clarifying their data in this revision. I recommend this paper for publication.

Reviewer #3 (Remarks to the Author):

After reviewing the resubmission my opinion remains the same, not to publish the manuscript. There are two main issues:

1) Most importantly, the authors have included some new data that they say supports their numerical studies that suggest chemotaxis. However, I don't see how the numbers work out for what they say to be consistent with the simulations. In particular, from fig 3e,f it seems that the drift velocity should be about 0.001-0.004 microns/s. In the experiments, the authors wait an hour (3600s), and say there is accumulation at the boundary due to chemotaxis. However, according to the simulations, the chemotactic drift in that time would only be 3.6-15 microns. The authors don't provide a size for the sample chamber but this seems inconsistent. Indeed in the simulated time of 5×10^4 s, Fig 3d and the text says diffusion dominates the motion rather than chemotactic drift.

The authors only say "cells are attracted to the region immediately adjacent to the agar interface", and then report differences in run times dependent on direction of travel up the gradient etc. Do they actually measure a chemotactic drift? Given the inconsistency above, I worry that another mechanism may have caused the attraction observed, especially since in an hour I'd expect that methionine would appreciably diffuse and may no longer maintain a gradient. Diffusive lengthscales of methionine would be certainly larger than the estimated chemotactic drift scale above (4-15 microns). Perhaps control experiments with no chemoattractant or mutants lacking chemotactic genes may shed light on this.

2) I find the use of "optimal" rather fuzzy and ill-defined. There is an idea that the cells balance power

output to chemotactic ability, but no real attempt to rigorously define or quantify this, especially as nutrient concentrations and/or competition are varied. This is a serious weakness since the title emphasizes this optimality.

Thornton et al., response to referees

Referee #1

We thank the reviewer for their constructive remarks and address their concerns in order.

“While the authors show that cell run duration depends on the direction along the gradient, they do not give the value for the chemotactic drift [...]”

We have now provided numbers for the chemotactic drift speed in experiments.

“Critically, given the substantial difference between the run duration up or down the gradient, one could expect the drift value to be rather high, on the order of several percent. This would be in contrast of the very small drift predicted by simulation (below 0.2 % in Fig. 3f). Understanding this difference between the simulations and experiments would be essential.”

Our new simulations do indeed predict drift speeds on the order of several percent (up to 11%), in line with experiments. This change came about due to a more extensive exploration of parameter space outlined in the newly extended supplementary information.

“The average run duration seems to increase in presence of a gradient, even for runs down the gradient, compared to medium without attractant. Is this a signalling or a metabolic effect?”

This is a good question, and one that we comment on in the manuscript and address quantitatively in the supplementary information. We cannot rule out a metabolic effect, though we note that the speed of cells is essentially the same in all regions of the gradient, and that cells maintain the same swimming speed throughout their swimming trajectories, which may last for 3-5 minutes each. We therefore consider that even though there may be some metabolic effect on the cells, there is no evidence of a chemokinetic effect, and so our model is still valid.

“[...] the arguments supporting the central claim of the manuscript, namely that haloarchaea have optimised energetics of chemotaxis in low-nutrient environments, have been largely removed from the manuscript. They are now replaced by a general argument that swimming at low speed is energetically less costly.”

We have updated the argument to re-emphasise that the archaea swim with optimised chemotactic energetics. We have explained our reasoning in more detail, outlining three ‘boundary conditions’ within which the cells must operate. Our new set of simulations, combined with experiments, show that the slow swimming is one part of the chemotactic strategy, along with run duration and maintaining linearity of response.

Referee #2

We are pleased that referee #2 is content with our previous submission. Given that the new pieces introduced do not concern their original areas of concern, we hope that the reviewer will remain satisfied.

Referee #3

We thank the reviewer for their comments on our manuscript and address their concerns below. The referee’s first criticism concerns the quantitative agreement between experiments and simulations:

"[...] I don't see how the numbers work out for what they say to be consistent with the simulations. In particular, from fig 3e,f it seems that the drift velocity should be about 0.001-0.004 microns/s. In the experiments, the authors wait an hour (3600s), and say there is accumulation at the boundary due to chemotaxis. However, according to the simulations, the chemotactic drift in that time would only be 3.6-15 microns. [...]"

We had previously estimated the chemotactic sensitivity conservatively, to align with other studies (e.g. ref. 38 in the main manuscript), which relate analytical theory to numerical simulations when studying *E. coli*. In constructing our revised simulations, we explored the effects of increasing chemotactic sensitivity in more depth, arriving at a configuration (described in methods and supplementary information) that matches the experiments much more closely while minimising response saturation. We have also given details of the sample chamber construction (methods). Quantitatively, our experimental drift velocity is now around 0.1 $\mu\text{m/s}$ (see Table 1 of the main manuscript), resulting in a drift of around 720 μm in an hour, a distance twice the size of our field of view in experiments.

"Diffusive lengthscales of methionine would be certainly larger than the estimated chemotactic drift scale above (4-15 microns). Perhaps control experiments with no chemoattractant or mutants lacking chemotactic genes may shed light on this."

The diffusive length scale of methionine in one dimension can be estimated as $l \sim \sqrt{2D_{meth}t}$, where D_{meth} is the molecular diffusivity of methionine. Using a value for D_{meth} of 600 $\mu\text{m}^2/\text{s}$ (the value for similarly sized molecules such as glucose), we obtain $l \sim 2$ mm which is approximately the distance from the agar at which we position of microscope field of view. This is now clarified in the methods and the main text. We also maintain that the runs that we observe perpendicular to the gradient are an effective control experiment, and note that the rest of the experiments in the paper and supplementary information – especially the controls in saltwater in the SI, which are the controls with no attractant that the referee describes – further support our conclusions.

"I find the use of "optimal" rather fuzzy and ill-defined. [...]"

This statement was weakened in response to the first round of criticisms, which we think may account for the resultant lack of definition in our results statement. Our new simulations give us renewed confidence in our results, and we have clearly outlined three boundaries within which the cells should operate and shown how they perform at, or very close to, the optimum chemotactic efficiency in this region, balancing the opposing demands of speed of gradient ascent against diffusion and hydrodynamic efficiency. Our simple linear chemotaxis model does not encompass temporally varying nutrient concentrations or competition between individuals – these are topics of study in their own right and outside the scope of the current work. Our cells are often found in the dilute limit in the wild, so we maintain that our simulations are already biologically relevant.

REVIEWER COMMENTS

Reviewer #1 (Remarks to the Author):

In this substantially revised version of the manuscript, the authors have satisfactorily addressed most of my concerns.

I have only few minor issues that need to be addressed/corrected before the manuscript is published:

- I am surprised with the $v_x \sim v_0$ dependence in the simulations. In this type of simulations, without rotational diffusion, one expects a scaling $v_x \sim v_0^2$ (e.g. ref. 37). I assume that the difference in scaling comes from rotational diffusion, and τ_{run} and D_r being close from each other. Could authors discuss that?

- In Fig. 3 h and g, it is not clear for which value of τ_{run} and D_r the simulations are carried out. In general, the value of the parameters should be made more explicit. It is not clear what they were for *E. coli* (except v_0) in particular.

- Looking at Fig 3i, the dependence of chemotactic efficiency on swimming velocity appears to be $\epsilon \sim v_0^{-1}$ and not $\epsilon \sim v_0^{-2}$ as stated on page 5.

- Authors' condition (ii) for efficient chemotaxis, namely that cells "...must be able to 'outpace' diffusing molecules by having a mean squared displacement that is greater than that for a diffusing molecule" only holds true for unstable gradients, doesn't it? This should be clarified. While many natural gradients are likely to be unstable, fairly stable environmental gradients (e.g., emerging when an attractant is slowly released from a source) certainly also exist.

Reviewer #3 (Remarks to the Author):

With the additional simulations, the authors have met my main concerns.

I think it is implied, but an explicit statement about whether in the saltwater and nutrient cases without the methionine gradient, there is any accumulation of cells similar to that observed near the agar and attributed to chemotaxis.

It seems that the chemotactic sensitivity (W) is much higher in the simulations than expected from other studies. The authors may want to compare and comment on how such a high sensitivity could be possible.

We would like to thank the reviewers for their time and assistance in improving the manuscript. We have addressed their concerns in order below; the reviewer's comments are in blue and our responses in black.

Reviewer #1 (Remarks to the Author):

In this substantially revised version of the manuscript, the authors have satisfactorily addressed most of my concerns.

I have only few minor issues that need to be addressed/corrected before the manuscript is published:

- I am surprised with the $v_x \sim v_0$ dependence in the simulations. In this type of simulations, without rotational diffusion, one expects a scaling $v_x \sim v_0^2$ (e.g. ref. 37). I assume that the difference in scaling comes from rotational diffusion, and τ_{run} and D_r being close from each other. Could authors discuss that?

We thank the reviewer for this observation, and we are happy to discuss it further in the context of the reference suggested. The scaling relation that the reviewer mentions is true in the 'weak chemotaxis' limit and must saturate at some point – v_x cannot grow quadratically with v_0 indefinitely, otherwise the cells would chemotax faster than they swim. Refs. 37 and 38 are both 'weak chemotaxis' models that observe a $v_x \sim v_0^2$ scaling, in the absence and presence (respectively) of rotational diffusion. The reviewer is quite right to point out that the rotational and run time scales are close to each other, and we have added to the discussion on p.5. Our simulations differ from those previous analytical and numerical studies in that (i) our run times and diffusional reorientation time scales are similar to each other (as the reviewer remarks) and (ii) our cells only reverse, they don't tumble through a fixed angle $\varphi \leq \pi/2$.

- In Fig. 3 h and g, it is not clear for which value of τ_{run} and D_r the simulations are carried out. In general, the value of the parameters should be made more explicit. It is not clear what they were for E. coli (except v_0) in particular.

The values for τ_{run} and D_r in Figs. 3h,g were those obtained in the experiments shown in Fig. 2f,i and Fig. 1f,h respectively. We have clarified this in the text (p4 & p5) and added more explicit detail in the methods section detailing the setup for our simulations.

- Looking at Fig 3i, the dependence of chemotactic efficiency on swimming velocity appears to be $\epsilon \sim v_0^{-1}$ and not $\epsilon \sim v_0^{-2}$ as stated on page 5.

We thank the reviewer for spotting these errors. They have now been corrected.

- Authors' condition (ii) for efficient chemotaxis, namely that cells "...must be able to 'outpace' diffusing molecules by having a mean squared displacement that is greater than that for a diffusing molecule" only holds true for unstable gradients, doesn't it? This should be clarified. While many natural gradients are likely to be unstable, fairly stable environmental gradients

(e.g., emerging when an attractant is slowly released from a source) certainly also exist.

We thank the reviewer for their suggestion, which has been incorporated into the text (p4).

Reviewer #3 (Remarks to the Author):

With the additional simulations, the authors have met my main concerns.

I think it is implied, but an explicit statement about whether in the saltwater and nutrient cases without the methionine gradient, there is any accumulation of cells similar to that observed near the agar and attributed to chemotaxis.

The text on p.3 has been amended accordingly.

It seems that the chemotactic sensitivity (W) is much higher in the simulations than expected from other studies. The authors may want to compare and comment on how such a high sensitivity could be possible.

We appreciate the reviewer's comment regarding the high chemotactic sensitivity we observed with halophilic archaea. We add a note of caution that although some components of the signal transduction system (e.g. the response regulator CheY) are conserved between eubacteria and archaea, there are distinct components as well, such as the archaeal-specific adapter protein CheF, which are only starting to be understood. Many of the best-known eubacterial signal transduction systems can modulate their response by (e.g.) methylation of receptors or by the dephosphorylation of CheY (CheZ performs this role in *E. coli*, for example). Moreover, receptors in eubacteria are clustered to allow for cooperative binding events as neighbouring receptors interact. Any of these modulation approaches could potentially be present in archaea but may not use eubacterial components. To emphasize the distinction between these archaea and other systems we have added a sentence and a reference in the final paragraph. Our work is attempting to characterize these differences, which do point to higher sensitivity. This makes sense given the nutrient-poor environs in which they live. We have expanded the discussion of the sensitivity parameter in the supplementary information and added words to the main text to indicate this (p.4).